# BAdd: Bias Mitigation through Bias Addition

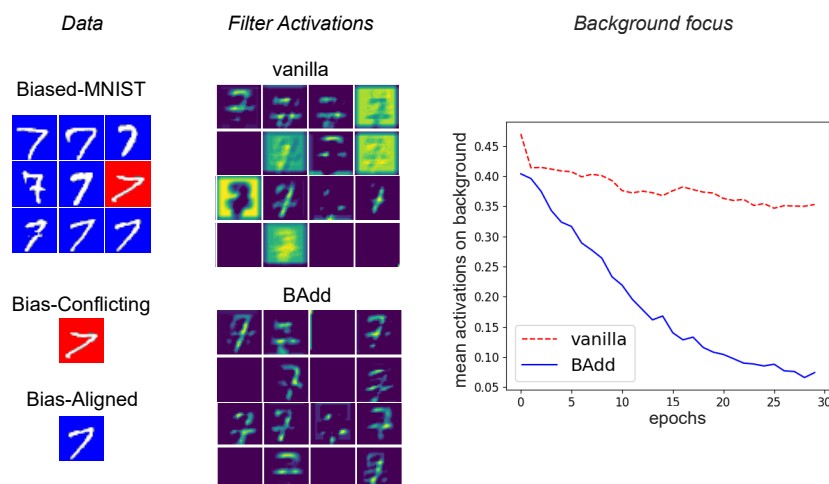

Figure 1: During training on Biased-MNIST, where the color-digit association is strong, a vanilla model struggles with bias, as reducing reliance on the protected attribute (here 'color') results in increased loss for samples that deviate from this spurious correlation. In contrast, BAdd results in learning bias-neutral feature representations of the digits, independent of color. This is evidenced by the activation maps on the samples where bias occurs.

## Abstract

Computer vision (CV) datasets often exhibit biases in the form of spurious correlations between certain attributes and target variables that are perpetuated by Deep Learning (DL) models. While recent efforts aim to mitigate such biases and foster bias-neutral representations, they fail in complex real-world scenarios. In particular, existing methods excel in controlled experiments on benchmarks with single-attribute injected biases, but struggle with complex multi-attribute biases that naturally occur in established CV datasets. Here, we introduce BAdd a simple yet effective method that allows for learning bias-neutral representations invariant to bias-inducing attributes. It achieves this by injecting features encoding these attributes into the training process. BAdd is evaluated on seven benchmarks and exhibits competitive performance, surpassing state-of-the-art methods on both single- and multi-attribute bias settings. Notably, it achieves +27.5% and +5.5% absolute accuracy improvements on the challenging multi-attribute benchmarks, FB-Biased-MNIST and CelebA, respectively.

## 1 Introduction

Deep Learning (DL) models have demonstrated impressive capabilities, as evidenced by ground-breaking performance across various Computer Vision (CV) tasks (Deng et al., 2019; Feichtenhofer et al., 2019; Tan et al., 2020). However, a concerning issue has emerged alongside these advancements: the potential for bias in AI systems, disproportionately impacting specific groups (Barocas et al., 2019; Fabbrizzi et al., 2022; Sarridis et al., 2023b). Specifically, when AI systems base their decisions - often indirectly - on attributes like age, gender, or race, they become discriminatory. Considering the profound impact AI decisions can have on individuals' lives, such biases should be mitigated prior to deployment in high-stakes applications (Bobadilla et al., 2013; Taigman

et al., 2014; Creswell et al., 2018; Tan et al., 2020). Moreover, even when such biases are not demographic-related but stem from "shortcuts" that prioritize irrelevant features, addressing them is crucial for building more robust and reliable CV systems (Sagawa et al., 2019; Li et al., 2023).

For CV systems, bias often originates from the composition of the datasets used for training (Fabbrizzi et al., 2022). One of the main ways in which bias arises in training sets is through a data selection process where specific groups of people or objects are largely associated with certain visual attributes (e.g., women are illustrated wearing earrings in the majority of images). When such data is used to train DL models, such common attribute associations can act as "shortcuts", leading the model to prioritize irrelevant attributes in its decision-making process (Zhang et al., 2018). Motivated by this issue, several approaches have been proposed to enable learning bias-neutral representations that are robust to the so-called *bias attributes* (Sarridis et al., 2023a; Hong & Yang, 2021; Barbano et al., 2022), i.e. attributes that exhibit spurious correlations with the target classes. Such methods often leverage labels associated with protected attributes to guide model training towards learning bias-neutral representations (Bahng et al., 2020; Cadene et al., 2019; Clark et al., 2019; Hong & Yang, 2021; Barbano et al., 2022; Sarridis et al., 2023a) through techniques like adversarial training (Kim et al., 2019; Wang et al., 2019) and regularization approaches (Tartaglione et al., 2021; Hong & Yang, 2021; Barbano et al., 2022; Sarridis et al., 2023a). A fundamental limitation of existing bias mitigation methods lies in their loss-based nature. Typically, such approaches introduce additional loss terms to penalize the biased model's behavior, which retroactively corrects bias that is already introduced in the model's learning process. While methods adopting this strategy may appear sound in theory and demonstrate state-of-the-art performance on simplistic datasets, they struggle with more complex forms of bias, especially when dealing with multiple biased attributes, and demonstrate sub-optimal performance. To overcome these challenges, there is a need for more proactive bias mitigation approaches that intervene earlier in the training process, addressing the root cause of bias propagation within the model itself. By disrupting the process through which bias is introduced to the model, we can build models that are more effective for a wide range of complex biases present in standard CV datasets.

In this paper, we recognize the need to enhance the applicability of bias-aware CV models in complex application settings by proposing BAdd, a simple yet effective method to mitigate bias at its core. The proposed method relies on the principle that injecting *bias-capturing features* into the penultimate layer's output enables learning representations invariant to these features (see Fig. 1). Deriving bias-capturing features is straightforward since it can be formulated as the task of predicting the values of biased attributes. BAdd intervenes in the mechanism by which bias is introduced to the DL models during training via the minimization of the loss function. In particular, a vanilla model optimizes its parameters by taking advantage of biases present in the data, as doing so reduces the overall loss. Such a model learns to prioritize features associated with the biased attributes, reinforcing and perpetuating the bias within its representations. To alleviate this issue, BAdd suggests that the intentional inclusion of bias-capturing features within the training process ensures that the attributes introducing the bias do not exert undue influence on the loss function optimization, and thus the trainable parameters of the model are not affected by them. In essence, BAdd decouples the learning of biased features from the optimization process and thus allows for learning bias-neutral representations. BAdd outperforms or is on par with state-of-the-art bias mitigation methods on a wide range of experiments involving four datasets with single attribute biases (i.e., Biased-MNIST, Biased-UTKFace, Waterbirds, and Corrupted-CIFAR10) and three datasets with multi-attribute biases (i.e., FB-Biased-MNIST, UrbanCars, and CelebA). Where BAdd shines is on datasets with multi-attribute biases, where it outperforms the state of the art by +27.5%, and +5.5% absolute accuracy improvements on FB-Biased-MNIST, and CelebA, respectively. In summary, the paper makes the following contributions: (i) we introduce BAdd, an effective methodology for learning bias-neutral representations concerning one or more protected attributes by incorporating bias-capturing features into the model's representations (ii) we provide an extensive evaluation involving seven benchmarks, demonstrating the superiority of BAdd on both single- and multi-attribute bias scenarios. The data and code are provided in supplementary material.

## 2 RELATED WORK

**Bias-aware image classification benchmarks.** Most standard benchmarks for evaluating bias mitigation techniques in CV involve artificially generated single-attribute biases. Biased-MNIST

(Bahng et al., 2020), a MNIST derivative dataset, associates each digit with a specific colored background. Similarly, Corrupted-CIFAR10 (Hendrycks & Dietterich, 2018) introduces biased textures across the classes of CIFAR10. The Waterbirds (Sagawa et al., 2019) dataset is constructed by cropping birds from the CUB-200 (Wah et al., 2011) dataset and transferring them onto backgrounds from the Places dataset (Zhou et al., 2017), introducing correlations between bird species and certain backgrounds (i.e., habitat types). On the other hand, datasets like Biased-UTKFace (Hong & Yang, 2021) and Biased-CelebA (Hong & Yang, 2021) are carefully selected subsets of UTKFace (Zhifei et al., 2017) and CelebA (Liu et al., 2015), respectively, designed to exhibit an association of 90% between specific attributes, such as `gender` and `race`. Despite their value in research, all these benchmarks share a crucial limitation: they are far from capturing the complexities of realistic scenarios, as they typically exhibit uniformly distributed single attribute biases. To approximate more realistic cases, recent works introduced benchmarks that involve multi-attribute biases, such as Biased-MNIST variations (Ahn et al., 2022; Shrestha et al., 2022a;b) and UrbanCars (Li et al., 2023). The latter introduces a multi-attribute bias setting by incorporating biases related to both background and co-occurring objects and the task is to classify the car body type into `urban` or `country` car. In addition to the above benchmarks, in this paper, we create a variation of Biased-MNIST, termed FB-Biased-MNIST, which builds on the background color bias in Biased-MNIST by injecting an additional foreground color bias. Furthermore, we consider a benchmark that utilizes the original, unmodified CelebA dataset but focuses on evaluating performance against the most prominent bias-inducing attributes in the dataset. This allows for evaluating bias-aware methods on multiple biases in a more realistic setting - without artificially enforced biases.

**Bias-aware approaches.** Efforts on learning bias-neutral representations using biased data encompass techniques like ensemble learning (Clark et al., 2019; Wang et al., 2020), contrastive learning (Hong & Yang, 2021; Barbano et al., 2022), adversarial frameworks (Xie et al., 2017; Alvi et al., 2018; Kim et al., 2019; Song et al., 2019; Wang et al., 2019; Adel et al., 2019), and regularization approaches (Cadene et al., 2019; Tartaglione et al., 2021; Hong & Yang, 2021; Sarridis et al., 2023a). For instance, the Learning Not to Learn (LNL) approach (Kim et al., 2019) penalizes models if they predict protected attributes, while the Domain-Independent (DI) approach (Wang et al., 2020) introduces the usage of domain-specific classifiers to mitigate bias. Entangling and Disentangling deep representations (EnD) (Tartaglione et al., 2021) suggests a regularization term that entangles or disentangles feature vectors w.r.t. their target and protected attribute labels. FairKL (Barbano et al., 2022) and BiasContrastive-BiasBalance (BC-BB) (Hong & Yang, 2021) are contrastive learning-based approaches that try to mitigate bias by utilizing the pairwise similarities of the samples in the feature space. Finally, there are several works that can be employed without utilizing the protected attribute labels, such as Learned-Mixin (LM) (Clark et al., 2019), Rubi (Cadene et al., 2019), Re-Bias (Bahng et al., 2020), Learning from Failure (LfF) (Nam et al., 2020), and FLAC (Sarridis et al., 2023a). The latter achieves state-of-the-art performance by utilizing a bias-capturing classifier and a sampling strategy that effectively focuses on the underrepresented groups. It is worth noting that methodologies for distributionally robust optimization (Sagawa et al., 2019; Liu et al., 2021a; Wu et al., 2023; Qiu et al., 2023; Li et al., 2022; 2023) are relevant to the field of bias mitigation, as they aim at mitigating biases arising from spurious correlations in the training data. Similarly to the aforementioned methods, Sagawa et al. (2019) and Li et al. (2022) suggest regularization terms to mitigate such correlations, while Liu et al. (2021a) and Wu et al. (2023) introduced methods that try to balance the datasets w.r.t. the spurious correlations by increasing or decreasing the weights of certain training samples. Based on the same idea, Qiu et al. (2023) focuses on reweighting the features rather than the samples. Finally, the Last Layer Ensemble (LLE) (Li et al., 2023) employs multiple augmentations to eliminate different biases (i.e., one type of augmentation for each type of bias). However, LLE requires extensive pre-processing (e.g., object segmentation), which makes it challenging or even infeasible to apply to new CV datasets. On the other hand, BAdd is a simple yet effective approach that can be easily applied to any network architecture and to any CV dataset.

## 3 METHODOLOGY

### 3.1 PROBLEM FORMULATION

Consider a dataset $\mathcal{D}$ comprising training samples $(\mathbf{x}^{(i)}, y^{(i)})$, where $\mathbf{x}^{(i)}$ represents the input sample and $y^{(i)}$ belongs to the set of target labels $\mathcal{Y}$. Let $h(\cdot)$ denote a model trained on $\mathcal{D}$ and $\mathbf{h}$ the model

feature representation (e.g., output of penultimate model layer). Let also $\mathcal{T}$ be the domain of tuples of *protected attributes*, e.g., $t = (male, 25, black) \in \mathcal{T}$ for protected attributes gender, age and race. The objective is to train $h$ such that the protected attributes are not used to predict the targets in $\mathcal{Y}$. In addition, we also assume that a bias-capturing model $b(\cdot)$, with feature representation $\mathbf{b}$, has been trained to predict the value of the protected attribute(s) $t \in \mathcal{T}$ from $\mathbf{x}$.

We define $\mathcal{D}$ as *biased* with respect to the protected attributes in $\mathcal{T}$ if there is high correlation of certain values in $\mathcal{Y}$ with a value or a combination of values of protected attributes in $\mathcal{T}$. Within a batch $\mathcal{B}$, samples exhibiting the dataset bias are termed *bias-aligned* ($\mathcal{B}_\mathcal{A}$), while those that deviate from it are referred to as *bias-conflicting* ($\mathcal{B}_\mathcal{C}$). The set $\mathcal{D}$ is assumed to include at least some bias-conflicting examples. Note that bias-aligned and bias-conflicting samples correspond to the over-represented and under-represented groups within $\mathcal{D}$, respectively. Using such a biased dataset for training often introduces model bias, by leading $h$ to encode information related to $t$. Our objective is to mitigate these dependencies between representations $\mathbf{h}$ and $\mathbf{b}$, leading to a bias-neutral feature representation.

## 3.2 THE VICIOUS CIRCLE OF BIAS

Training a classification model $h(\cdot)$ on a biased dataset $\mathcal{D}$ very often prioritizes learning features related to the protected attributes instead of features directly characterizing the target class. This phenomenon arises in cases of high correlation between protected attributes and targets, provided that the protected attribute's visual characteristics are easier to capture than the visual characteristics of the target (Zhang et al., 2018). Below, we delve into the details behind a vanilla model's inherent inclination towards this kind of bias and explain how the proposed approach addresses this limitation.

First, let us consider the Cross-Entropy loss on a batch of samples $\mathcal{B} = \mathcal{B}_\mathcal{A} \cup \mathcal{B}_\mathcal{C}$:

$$\mathcal{L} = -\frac{1}{N} \sum_{i \in \mathcal{B}} \sum_{k=1}^{K} y_k^{(i)} \log \hat{y}_k^{(i)} = -\frac{1}{N} \sum_{i \in \mathcal{B}_\mathcal{A}} \sum_{k=1}^{K} y_k^{(i)} \log \hat{y}_k^{(i)} - \frac{1}{N} \sum_{i \in \mathcal{B}_\mathcal{C}} \sum_{k=1}^{K} y_k^{(i)} \log \hat{y}_k^{(i)} = \mathcal{L}_{\mathcal{B}_\mathcal{A}} + \mathcal{L}_{\mathcal{B}_\mathcal{C}},$$

(1)

where $N$ is the number of samples within a batch, and $K$ the number of target classes. The predictions $\hat{y}_k^{(j)}$ are computed via multinomial logistic regression, as follows:

$$\hat{y}_k^{(j)} = \sigma_k(\mathbf{z}(\mathbf{x}^{(j)}; \boldsymbol{\theta_h})),$$

(2)

where $j$ is the index of input sample $\mathbf{x}^{(j)}$, $\boldsymbol{\theta_h}$ the learnable parameters of model $h(\cdot)$ and $\sigma_k$ the $k$-th class probability after applying the softmax function on the logits $\mathbf{z}(\mathbf{x}^{(j)}; \boldsymbol{\theta_h})$.

Given that $||\mathcal{B}_\mathcal{A}|| >> ||\mathcal{B}_\mathcal{C}||$, we can assume that there exists a point in the training process at which the model has learned to be accurate on the bias-aligned samples $\mathcal{B}_\mathcal{A}$ misguidedly relying on protected attributes' features, so that $\mathcal{L}_{\mathcal{B}_\mathcal{A}} \approx 0$, while at the same time $\mathcal{L}_{\mathcal{B}_\mathcal{C}} >> 0$. Consequently, backpropagating the gradients of $\mathcal{L}$ will update the parameters $\boldsymbol{\theta_h}$ in a way that steers the model towards accurately predicting the bias-conflicting samples $\mathcal{B}_\mathcal{C}$ in order to further reduce $\mathcal{L}$, which stops reliance of $h(\cdot)$ on the protected attributes. The major limitation of a vanilla model is directly connected to the loss behavior when the model processes the next mini-batch. In particular, the step the model makes towards correctly predicting the samples in $\mathcal{B}_\mathcal{C}$, thus reducing $\mathcal{L}_{\mathcal{B}_\mathcal{C}}$, adversely affects the loss w.r.t. the bias-aligned samples, which is now $\mathcal{L}_{\mathcal{B}_\mathcal{A}} >> 0$, as $h(\cdot)$ relies less on the protected attributes and at the same time it is impossible to learn to encode the target with only one batch of $||\mathcal{B}_\mathcal{C}||$ bias-conflicting samples. This leads to a loss spike for the bias-aligned samples that in the next iteration will restore the model's parameters $\boldsymbol{\theta_h}$ to their initial state (encoding the protected attributes' features) in order to again achieve a much lower $\mathcal{L}$. Figure 2 illustrates this behavior through a snapshot of the losses and the gradients related to the bias-aligned and bias-conflicting samples during several training steps of the vanilla model (refer to the Appendix for the BAdd model behavior). In this example, to emphasize the described phenomenon, we primarily include bias-aligned samples, with only 2 batches of bias-conflicting samples introduced every 200 training steps.

To better expose this behavior, let us consider the derivative of the loss of equation 1 with respect to a parameter $\theta_h^0$ for the $i$-th sample:

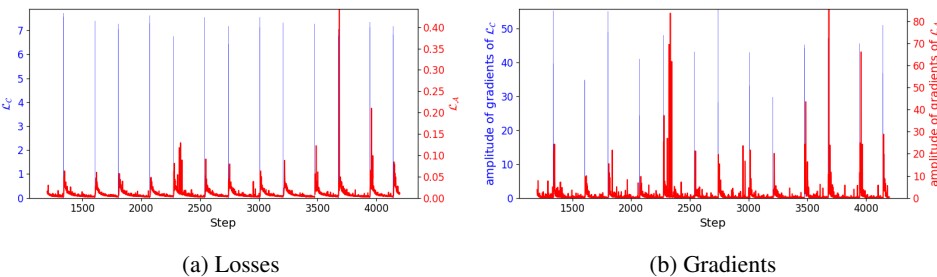

(a) Losses                                        (b) Gradients

Figure 2: Biased-MNIST bias-conflicting samples trigger spikes on $\mathcal{L}_{\mathcal{A}}$ and gradients of $\mathcal{L}_{\mathcal{A}}$. Blue bars indicate the steps where bias-conflicting samples occur, with height representing $y$-axis values.

$$\frac{\partial \mathcal{L}^{(i)}}{\partial \theta_h^0} = y_\kappa \frac{\partial \log \sigma_\kappa(\mathbf{z}(\mathbf{x}^{(i)}; \boldsymbol{\theta_h}))}{\partial \theta_h^0} = y_\kappa \frac{1}{\sigma_\kappa(\mathbf{z}(\mathbf{x}^{(i)}; \boldsymbol{\theta_h}))} \frac{\partial \sigma_\kappa(\mathbf{z}(\mathbf{x}^{(i)}; \boldsymbol{\theta_h}))}{\partial \theta_h^0}, \tag{3}$$

where $\kappa$ is the correct class, according to the ground truth (i.e., $y_\kappa = 1$). Setting $A_0^{(i)} = \frac{\partial \sigma_\kappa(\mathbf{z}(\mathbf{x}^{(i)}; \boldsymbol{\theta_h}))}{\partial \theta_h^0}$ and $\sigma_\kappa^{(i)} = \sigma_\kappa(\mathbf{z}(\mathbf{x}^{(i)}; \boldsymbol{\theta_h}))$, the derivative for a batch becomes

$$\frac{\partial \mathcal{L}}{\partial \theta_h^0} = -\frac{1}{N} \Big( \sum_{i:\mathbf{x}^{(i)} \in \mathcal{B}_{\mathcal{A}}} \frac{1}{\sigma_\kappa^{(i)}} A_0^{(i)} + \sum_{j:\mathbf{x}^{(j)} \in \mathcal{B}_{\mathcal{C}}} \frac{1}{\sigma_\kappa^{(j)}} A_0^{(j)} \Big). \tag{4}$$

After the model has learned to predict the targets based on the protected attributes, $\sigma_\kappa^{(i)}$ is large (close to 1) while $A_0^{(i)}$ is small, as $h(\cdot)$ already correctly predicts samples in $\mathcal{B}_{\mathcal{A}}$. In contrast, $\sigma_\kappa^{(j)}$ is small while $A_0^{(j)}$ is large. The model update therefore strongly depends on the samples in $\mathcal{B}_{\mathcal{C}}$. After the update step, however, $\sigma_\kappa^{(i)}$ becomes smaller, $A_0^{(i)}$ becomes larger and given that $||\mathcal{B}_{\mathcal{A}}|| >> ||\mathcal{B}_{\mathcal{C}}||$, the derivative is now dominated by samples in $\mathcal{B}_{\mathcal{A}}$, and the parameters revert back to their previous values. In other words, any progress the model makes towards reducing its bias is counteracted by the loss function, which is lower when the model focuses on the easier-to-learn, biased samples. This essentially traps the model in a vicious circle where the model is condemned to encode the protected attributes instead of the targets.

### 3.3  BADD

BAdd proposes incorporating the features $\mathbf{b}$ that capture the protected attributes of the dataset in the model's feature representation $\mathbf{h}$. Feature representation $\mathbf{b}$ encapsulates all the desired protected attributes and can be considered as $\mathbf{b} = \mathbf{b}_1 + \mathbf{b}_2 + \cdots + \mathbf{b}_M$ where $M = |\mathcal{T}|$ is the number of protected attributes in the dataset. These features can be obtained either by training a bias-capturing classifier or, in case the protected attribute labels are known, by projecting them into the dimension of $\mathbf{h}$ through one-hot encoding. In the first case, a typical DL model is trained to predict the attribute of interest, e.g., race, gender, hair color, or background, which the main model should avoid "using" in its prediction. Note that training a classifier to predict the protected attributes encourages the learning of richer, more diverse latent features associated with them. This approach helps capture subtle, underlying patterns in the data that may otherwise be lost when relying solely on labeled attributes. On the other hand, directly projecting attribute labels through one-hot encoding is easier to implement and computationally less intensive. However, it may not capture the complexity of visual features as effectively as a dedicated bias-capturing classifier.

The combined representation $\mathbf{h} + \mathbf{b}$ is then fed to the final classification layer. Thus, during training, model predictions are computed as $\hat{y}_k^{(j)} = \sigma_k(\mathbf{W}(\mathbf{h}(\mathbf{x}^{(j)}; \boldsymbol{\theta_h}) + \mathbf{b}(\mathbf{x}^{(j)})) + \boldsymbol{\rho})$, where $\mathbf{W}$ and $\boldsymbol{\rho}$ are the parameters of the last linear layer of $h(\cdot)$. By incorporating the biased features $\mathbf{b}$ into the training, we equip the model with the necessary information to consistently account for the bias-aligned samples. This means that the $\mathcal{L}_{\mathcal{B}_{\mathcal{A}}}$ values are consistently close to 0, preventing the loss spikes, and thus enabling features $\mathbf{h}$ to encode information about the target classes rather than the protected attributes, without having a negative impact on the loss of the bias-aligned samples. In

terms of the training process implied by equation 4, the addition of $\mathbf{b}$ entails invariably large $\sigma_\kappa^{(i)}$ and small $A_0^{(i)}$, thus forcing model updates to depend on the samples of $\mathcal{B}_\mathcal{C}$ consequently eliminating the effect of bias-aligned samples. Having learned a bias-neutral representation $\mathbf{h}$, a final fine-tuning step is required to account for the fact that $\mathbf{b}$ will not be added to input samples at inference time. During this fine-tuning stage, only the final classification layer (i.e., $\mathbf{W}$ and $\boldsymbol{\rho}$) is updated using $\mathbf{h}$ as input. After this step, model predictions are computed using $\hat{y}_k^{(j)} = \sigma_k(\mathbf{W}\mathbf{h}(\mathbf{x}^{(j)}; \boldsymbol{\theta_h}) + \boldsymbol{\rho})$.

While BAdd is found to be very effective in mitigating bias in cases of highly biased datasets, we observe that it does not adversely affect model performance in cases of datasets where bias is much less prevalent (cf. experimental results in the Appendix). This is an expected behavior because, in low- or no-bias scenarios, the bias-capturing features, $\mathbf{b}$, do not contain information that the model can exploit to predict the target variables. As a result, these features act as noise, which the model naturally learns to ignore without affecting its overall performance.

## 4 EXPERIMENTAL SETUP

### 4.1 DATASETS

Biased-MNIST (Bahng et al., 2020) is an MNIST derivative dataset (LeCun, 1998) that serves as a benchmark for bias mitigation methods. It features digits with colored backgrounds, introducing bias through the association of each digit with a specific color. The degree of bias, represented by the probability $q$ of samples belonging to class $y$ and at the same time possessing the attributes $t$, thus determining the strength of this spurious correlation. We consider four variations of Biased-MNIST with $q$ values of 0.99, 0.995, 0.997, and 0.999, as commonly used in previous works. Biased-CelebA (Hong & Yang, 2021) is a subset of the CelebA facial image dataset, which is annotated with 40 binary attributes. Biased-CelebA considers `gender` as the target, while `HeavyMakeup` and `WearingLipstick` serve as the attributes introducing bias. Similarly, Biased-UTKFace (Hong & Yang, 2021) is a subset of the facial image UTKFace dataset that is annotated with `gender`, `race`, and `age` labels. `Gender` is the target label, with `race` or `age` considered as protected attributes. In both Biased-CelebA and Biased-UTKFace, the enforced correlation between the target and protected attributes is 0.9. The Corrupted-CIFAR10 dataset (Hendrycks & Dietterich, 2018) consists of 10 classes with texture-related biases uniformly distributed in the training data using four different values of $q$: 0.95, 0.98, 0.99, and 0.995. Finally, the Waterbirds (Sagawa et al., 2019) dataset demonstrates a co-occurrence of 0.95 between waterbirds (or landbirds) and aquatic environments (or terrestrial environments) as background.

Table 1: Fairness of a vanilla `gender` classifier trained on default CelebA w.r.t. potentially biased attributes. Accuracy for the under-represented groups (e.g., `male-WearingLipstick`) is denoted as "Bias-Conflicting" and the average accuracy across all the subgroups defined by the `gender` and the attribute is denoted as "Unbiased".

Table 2: CelebA: co-occurrence between `gender` and `WearingLipstick` and `HeavyMakeup` attributes.

| Attribute | Accuracy | |
|---|---|---|
| | Unbiased | Bias-conflicting |
| Smiling | 98.6 | 98.5 |
| WearingNecklace | 98.1 | 97.3 |
| WearingEarrings | 97.7 | 96.3 |
| BlondHair | 96.9 | 94.9 |
| Eyeglasses | 96.5 | 94.5 |
| WearingLipstick | 95.2 | 91.1 |
| HeavyMakeup | 93.0 | 86.7 |

| Attribute | Co-occurrence | |
|---|---|---|
| | Females | Males |
| WearingLipstick | 80.6% | 0.06% |
| HeavyMakeup | 66.3% | 0.03% |

Similar to the Biased-MNIST, we create FB-Biased-MNIST, an extension that enhances the bias introduced by the background color in Biased-MNIST, by injecting foreground color bias into the

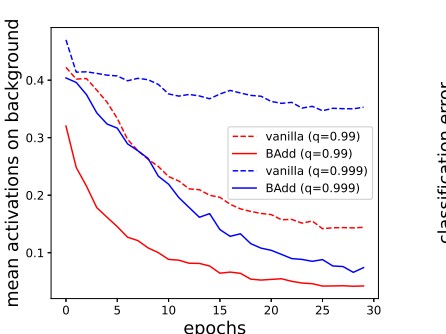 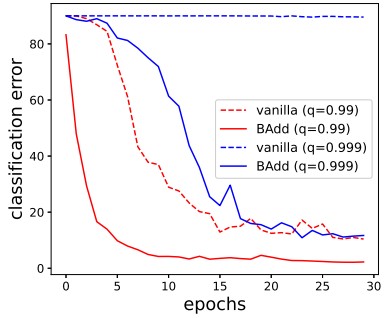

(a) Mean activation values of the first convolutional layer on sample backgrounds.

(b) Classification error during the first 30 training epochs.

Figure 3: Vanilla vs BAdd: Mean biased filter activation values and classification error.

dataset. Considering the increased complexity of this dataset compared to Biased-MNIST, we opt for lower $q$ values, namely 0.9, 0.95, and 0.99. Furthermore, the UrbanCars dataset is a synthetic dataset that exhibits a 0.95 co-occurrence between car body type and the background and/or certain objects relevant to urban or rural regions. We also assess the performance of bias mitigation methods on the default CelebA dataset (Liu et al., 2015) that is devoid of injected biases. To properly select attributes with a measurable degree of bias that could lead to problematic model behavior, we consider the performance disparities of a standard `gender` classifier trained on CelebA with respect to various potentially biased attributes. Subsequently, we identify the top two attributes (`WearingLipstick`, `HeavyMakeup`) with the most significant impact on the model's performance (Tab. 1), as a result of the strong association between these attributes and females (Tab. 2).

## 4.2 EVALUATION PROTOCOL

Different evaluation setups are used for each dataset, following the conventions of the literature to be comparable with previous works. In particular, following the Hong & Yang (2021); Barbano et al. (2022); Sarridis et al. (2023a), the test sets used for Biased-MNIST and FB-Biased-MNIST are composed using $q = 0.1$ that ensures each digit-color group is equally represented. For Biased-UTKFace and CelebA datasets, we utilize bias-conflicting and unbiased accuracy as in (Sarridis et al., 2023a; Hong & Yang, 2021). In particular, bias-conflicting accuracy refers to the accuracy of the under-represented samples (e.g., males wearing lipstick), and unbiased accuracy refers to the average accuracy across all the subgroups defined by the target (i.e., `gender`) and the protected attributes (i.e.,`WearingLipstick` and `HeavyMakeup`). The original test set, as shared by the dataset creators, is used in the case of Corrupted-CIFAR10. Regarding the Waterbirds dataset, we employ the average accuracy between different groups and the Worst-Group (WG) accuracy. Finally, for the UrbanCars dataset, we measure the In Distribution Accuracy (I.D. Acc) which is the weighted average accuracy w.r.t. the different groups, where the correlation ratios are the weights. The I.D. Acc is used as a baseline to measure the accuracy drop with respect to the background (BG Gap), co-occurring objects (CoObj Gap), and both the background and co-occurring objects (BG+CoObj Gap). Note that the implementation details are provided in the Appendix.

## 5 RESULTS

### 5.1 SINGLE ATTRIBUTE BIAS

Table 3 presents the performance of BAdd against nine competing methods. The proposed approach consistently surpasses state-of-the-art, demonstrating accuracy improvements ranging from 0.1% to 0.8% across different $q$ values. Fig. 3 illustrates the mean activations in image regions where bias occurs alongside the corresponding classification errors for both the Vanilla and BAdd approaches. This makes clear that the proposed method effectively reduces activations in areas where bias appears, leading to significant improvements in classification performance. This is particularly

Table 3: Evaluation on Biased-MNIST for different bias levels.

| Method | $q$ | | | |
|---|---|---|---|---|
| | 0.99 | 0.995 | 0.997 | 0.999 |
| Vanilla | $90.8_{\pm 0.3}$ | $79.5_{\pm 0.1}$ | $62.5_{\pm 2.9}$ | $11.8_{\pm 0.7}$ |
| LM (Clark et al., 2019) | $91.5_{\pm 0.4}$ | $80.9_{\pm 0.9}$ | $56.0_{\pm 4.3}$ | $10.5_{\pm 0.6}$ |
| Rubi (Cadene et al., 2019) | $85.9_{\pm 0.1}$ | $71.8_{\pm 0.5}$ | $49.6_{\pm 1.5}$ | $10.6_{\pm 0.5}$ |
| ReBias (Bahng et al., 2020) | $88.4_{\pm 0.6}$ | $75.4_{\pm 1.0}$ | $65.8_{\pm 0.3}$ | $26.5_{\pm 1.4}$ |
| LfF (Nam et al., 2020) | $95.1_{\pm 0.1}$ | $90.3_{\pm 1.4}$ | $63.7_{\pm 20.3}$ | $15.3_{\pm 2.9}$ |
| LNL (Kim et al., 2019) | $86.0_{\pm 0.2}$ | $72.5_{\pm 0.9}$ | $57.2_{\pm 2.2}$ | $18.2_{\pm 1.2}$ |
| EnD (Tartaglione et al., 2021) | $94.8_{\pm 0.3}$ | $94.0_{\pm 0.6}$ | $82.7_{\pm 0.3}$ | $59.5_{\pm 2.3}$ |
| BC-BB (Hong & Yang, 2021) | $95.0_{\pm 0.9}$ | $88.2_{\pm 2.3}$ | $82.8_{\pm 4.2}$ | $30.3_{\pm 11.1}$ |
| FairKL (Barbano et al., 2022) | $97.9_{\pm 0.0}$ | $97.0_{\pm 0.0}$ | $96.2_{\pm 0.2}$ | $90.5_{\pm 1.5}$ |
| FLAC (Sarridis et al., 2023a) | $97.9_{\pm 0.1}$ | $96.8_{\pm 0.0}$ | $95.8_{\pm 0.2}$ | $89.4_{\pm 0.8}$ |
| BAdd | $\mathbf{98.1}_{\pm \mathbf{0.2}}$ | $\mathbf{97.3}_{\pm \mathbf{0.2}}$ | $\mathbf{96.3}_{\pm \mathbf{0.2}}$ | $\mathbf{91.7}_{\pm \mathbf{0.6}}$ |

Table 4: Mean pairwise cosine similarity between 10 variations of each Biased-MNIST test sample, where each sample variation has a different background color.

| Method | $q$ | | | |
|---|---|---|---|---|
| | 0.99 | 0.995 | 0.997 | 0.999 |
| Vanilla | 0.889 | 0.854 | 0.811 | 0.416 |
| BAdd | 0.985 | 0.985 | 0.980 | 0.973 |

pronounced in experiments with $q = 0.999$, where the vanilla approach struggles with the impact of the biased attribute. Furthermore, the efficacy of BAdd to learn feature representations that are independent of the protected attribute is illustrated in Tab. 4. Specifically, Tab. 4 shows the mean pairwise cosine similarity between 10 variations of each Biased-MNIST test sample, where each variation has a different background color. BAdd leads to similarity values consistently close to 1 for all correlation ratios, which is not the case for the vanilla model that cannot maintain high similarities when the correlation ratio increases (e.g., 0.416 similarity for $q = 0.999$).

Table 5 illustrates the performance comparison of BAdd against the competing methods on the Biased-UTKFace dataset, where `race` and `age` are considered as protected attributes. Across both protected attributes, the proposed approach outperforms competing methods on bias-conflicting samples, achieving improvements of +1.1% (`race`) and +1.9% (`age`) compared with the second best. In terms of unbiased performance, BAdd exhibits only marginal differences compared to the state-of-the-art methods, with increases of 0.2% (`race`) and decreases of 0.3% (`age`).

Table 5: Evaluation of the proposed method on Biased-UTKFace for two different protected attributes, namely `race` and `age`, with `gender` as the target attribute.

| Method | Bias | | | |
|---|---|---|---|---|
| | Race | | Age | |
| | Unbiased | Bias-conflicting | Unbiased | Bias-conflicting |
| Vanilla | $87.4_{\pm 0.3}$ | $79.1_{\pm 0.3}$ | $72.3_{\pm 0.3}$ | $46.5_{\pm 0.2}$ |
| LNL (Kim et al., 2019) | $87.3_{\pm 0.3}$ | $78.8_{\pm 0.6}$ | $72.9_{\pm 0.1}$ | $47.0_{\pm 0.1}$ |
| EnD (Tartaglione et al., 2021) | $88.4_{\pm 0.3}$ | $81.6_{\pm 0.3}$ | $73.2_{\pm 0.3}$ | $47.9_{\pm 0.6}$ |
| BC-BB (Hong & Yang, 2021) | $91.0_{\pm 0.2}$ | $89.2_{\pm 0.1}$ | $79.1_{\pm 0.3}$ | $71.7_{\pm 0.8}$ |
| FairKL (Barbano et al., 2022) | $85.5_{\pm 0.7}$ | $80.4_{\pm 1.0}$ | $72.7_{\pm 0.2}$ | $48.6_{\pm 0.6}$ |
| FLAC (Sarridis et al., 2023a) | $92.0_{\pm 0.2}$ | $92.2_{\pm 0.7}$ | $\mathbf{80.6}_{\pm \mathbf{0.7}}$ | $71.6_{\pm 2.6}$ |
| BAdd | $\mathbf{92.2}_{\pm \mathbf{0.2}}$ | $\mathbf{93.3}_{\pm \mathbf{0.2}}$ | $80.3_{\pm 0.8}$ | $\mathbf{73.6}_{\pm \mathbf{1.0}}$ |

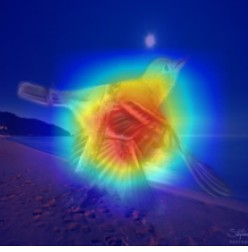 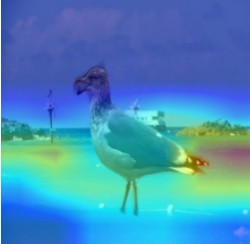 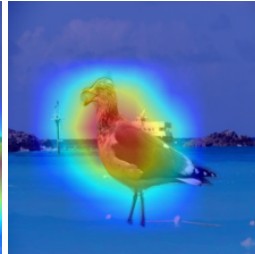

| (a) Method: Vanilla; Sample: bias-conflicting | (b) Method: BAdd; Sample: bias-conflicting | (c) Method: Vanilla; Sample: bias-aligned | (d) Method: BAdd Samlple: bias-aligned |

Figure 4: Vanilla vs BAdd: GradCam activations on bias-aligned (waterbird with sea background) and bias-conflicting (land bird with sea background) samples of Waterbirds dataset.

In the final single-attribute evaluation scenario, biases stemming from image background or textures are considered. As for the texture biases, the results obtained on the Corrupted-CIFAR10 dataset for four different bias ratios are summarized in Tab. 6. Given the complexity of training a bias-capturing classifier in this scenario, BAdd is implemented using a projection of one-hot vectors representing the texture labels to the feature space of the main model. Notably, BAdd consistently outperforms state-of-the-art across all Corrupted-CIFAR10 variations. Specifically, it achieves improvements of 6.5%, 3.1%, 3.4%, and 1.6% for correlation ratios of 0.95, 0.98, 0.99, and 0.995, respectively. Table 7, demonstrates the performance of BAdd on the Waterbirds dataset compared to the state-of-the-art methods for distributionally robust optimization. Here, BAdd reaches the state-of-the-art WG accuracy, i.e., 92.9%, and demonstrates competitive average accuracy, i.e., 93.6%. To further illustrate the effect of BAdd on the behavior of $h(\cdot)$, we visualize GradCam (Selvaraju et al., 2017) activations for a bias-aligned and a bias-conflicting sample of Waterbirds in Fig. 4. As can be easily noticed, the model trained with BAdd effectively focuses on birds, remaining unaffected by the presence of biases (i.e., background). In contrast, the vanilla model relies primarily on the background for its predictions.

Table 6: Evaluation on Corrupted-CIFAR10.

| Method | $q$ | | | |
|---|---|---|---|---|
| | 0.95 | 0.98 | 0.99 | 0.995 |
| Vanilla | $39.4_{\pm 0.6}$ | $30.1_{\pm 0.7}$ | $25.8_{\pm 0.3}$ | $23.1_{\pm 1.2}$ |
| EnD (Tartaglione et al., 2021) | $36.6_{\pm 4.0}$ | $34.1_{\pm 4.8}$ | $23.1_{\pm 1.1}$ | $19.4_{\pm 1.4}$ |
| ReBias (Bahng et al., 2020) | $43.4_{\pm 0.4}$ | $31.7_{\pm 0.4}$ | $25.7_{\pm 0.2}$ | $22.3_{\pm 0.4}$ |
| LfF (Nam et al., 2020) | $50.3_{\pm 1.6}$ | $39.9_{\pm 0.3}$ | $33.1_{\pm 0.8}$ | $28.6_{\pm 1.3}$ |
| FairKL (Barbano et al., 2022) | $50.7_{\pm 0.9}$ | $41.5_{\pm 0.4}$ | $36.5_{\pm 0.4}$ | $33.3_{\pm 0.4}$ |
| FLAC (Sarridis et al., 2023a) | $53.0_{\pm 0.7}$ | $46.0_{\pm 0.2}$ | $39.3_{\pm 0.4}$ | $34.1_{\pm 0.5}$ |
| BAdd | $\mathbf{59.5_{\pm 0.5}}$ | $\mathbf{49.1_{\pm 0.3}}$ | $\mathbf{42.7_{\pm 0.2}}$ | $\mathbf{35.7_{\pm 0.6}}$ |

Table 7: Evaluation on Waterbirds.

| Method | WG Acc. | Avg. Acc. |
|---|---|---|
| JTT (Liu et al., 2021a) | $86.7_{\pm 1.5}$ | $93.3_{\pm 0.3}$ |
| DISC (Wu et al., 2023) | $88.7_{\pm 0.4}$ | $93.8_{\pm 0.7}$ |
| GroupDro (Sagawa et al., 2019) | $90.6_{\pm 1.1}$ | $91.8_{\pm 0.3}$ |
| DFR (Kirichenko et al., 2022) | $\mathbf{92.9_{\pm 0.2}}$ | $\mathbf{94.2_{\pm 0.4}}$ |
| BAdd | $\mathbf{92.9_{\pm 0.3}}$ | $93.6_{\pm 0.2}$ |

Table 8: Evaluation on FB-Biased-MNIST.

| Method | $q$ | | |
|---|---|---|---|
| | 0.9 | 0.95 | 0.99 |
| Vanilla | $82.5_{\pm 0.8}$ | $57.9_{\pm 1.7}$ | $25.5_{\pm 0.6}$ |
| EnD (Tartaglione et al., 2021) | $82.5_{\pm 1.0}$ | $57.5_{\pm 2.0}$ | $25.7_{\pm 0.8}$ |
| BC-BB (Hong & Yang, 2021) | $80.9_{\pm 2.4}$ | $66.0_{\pm 2.4}$ | $40.9_{\pm 3.4}$ |
| FairKL (Barbano et al., 2022) | $87.6_{\pm 0.8}$ | $61.6_{\pm 2.6}$ | $42.0_{\pm 1.1}$ |
| FLAC (Sarridis et al., 2023a) | $84.4_{\pm 0.8}$ | $63.1_{\pm 1.7}$ | $32.4_{\pm 1.1}$ |
| BAdd | $\mathbf{95.6_{\pm 0.3}}$ | $\mathbf{89.0_{\pm 1.8}}$ | $\mathbf{69.5_{\pm 2.5}}$ |

## 5.2 MULTI-ATTRIBUTE BIAS

As previously discussed, evaluating bias mitigation performance solely in single-attribute scenarios provides an initial assessment but fails to capture the complexities of real-world settings. In this section, we present the performance of BAdd in two multi-attribute bias evaluation setups, namely on FB-Biased-MNIST and CelebA datasets. As depicted in Tab. 8, competing methods struggle to effectively mitigate bias on the FB-Biased-MNIST dataset, while BAdd consistently outperforms the second-best performing methods by significant margins of 8%, 23%, and 27.5% for $q$ of 0.9, 0.95, and 0.99, respectively. Notably, even in an artificial dataset like FB-Biased-MNIST, existing approaches struggle to address multiple biases. Table 9 demonstrates the performance of BAdd on UrbanCars, a dataset with artificially injected bias that is much more challenging than FB-Biased-MNIST. As observed, most compared methods struggle to address both the background and the co-occurring object biases. The only exception is LLE, which employs architectural modifications and specific bias-oriented augmentations to tackle each type of bias. However, it should be stressed that this approach requires extensive pre-processing, including object segmentation, making its application to other CV datasets very effort-intensive or even infeasible. Finally, as an example of a real-world dataset without artificially injected biases, we use the default CelebA dataset, where `gender` is the target attribute and multiple biases are present. As shown in Tab. 10, BAdd consistently improves performance for the attributes introducing bias, achieving absolute accuracy improvements of +3.5% and +5.5% for the bias-conflicting samples and +1.1% and +2.1% average accuracy across the subgroups compared to the second-best performing methods.

Table 9: Evaluation on UrbanCars.

| Method | I.D. Acc | BG Gap | CoObj Gap | BG+CoObj Gap |
|---|---|---|---|---|
| LfF (Nam et al., 2020) | 97.2 | -11.6 | -18.4 | -63.2 |
| JTT (Liu et al., 2021a) | 95.9 | -8.1 | -13.3 | -40.1 |
| Debian (Li et al., 2022) | 98.0 | -14.9 | -10.5 | -69.0 |
| GroupDro (Sagawa et al., 2019) | 91.6 | -10.9 | -3.6 | -16.4 |
| DFR (Kirichenko et al., 2022) | 89.7 | -10.7 | -6.9 | -45.2 |
| LLE (Li et al., 2023) | 96.7 | **-2.1** | -2.7 | -5.9 |
| BAdd | $91.0_{\pm0.7}$ | $-4.3_{\pm0.4}$ | $\mathbf{-1.6_{\pm1.0}}$ | $\mathbf{-3.9_{\pm0.4}}$ |

Table 10: Evaluation of the proposed method on CelebA for multiple attributes introducing bias, namely `WearingLipstick` and `HeavyMakeup`. `Gender` is the target attribute.

| Method | Biases | | | |
|---|---|---|---|---|
| | WearingLipstick | | HeavyMakeup | |
| | Unbiased | Bias-conflicting | Unbiased | Bias-conflicting |
| Vanilla | $95.2_{\pm0.3}$ | $91.1_{\pm0.6}$ | $93.0_{\pm0.8}$ | $86.7_{\pm1.6}$ |
| EnD (Tartaglione et al., 2021) | $95.1_{\pm0.4}$ | $91.0_{\pm0.7}$ | $92.3_{\pm0.7}$ | $85.3_{\pm1.5}$ |
| BC-BB (Hong & Yang, 2021) | $91.6_{\pm2.6}$ | $85.8_{\pm5.1}$ | $89.7_{\pm2.3}$ | $81.8_{\pm4.5}$ |
| FairKL (Barbano et al., 2022) | $82.7_{\pm0.4}$ | $74.7_{\pm0.3}$ | $84.4_{\pm0.9}$ | $77.9_{\pm1.2}$ |
| FLAC (Sarridis et al., 2023a) | $95.4_{\pm0.3}$ | $91.6_{\pm0.5}$ | $93.2_{\pm0.3}$ | $87.2_{\pm0.7}$ |
| BAdd | $\mathbf{96.5_{\pm0.2}}$ | $\mathbf{95.1_{\pm0.4}}$ | $\mathbf{95.3_{\pm0.5}}$ | $\mathbf{92.7_{\pm1.1}}$ |

## 6 CONCLUSION

In this work, we propose a method for bias mitigation in CV deep-learning models, termed BAdd. The proposed method injects bias-capturing features in the features of a model in order to force the model parameter updates to rely only on unbiased samples, thus leading to bias-neutral representations. The main requirement for BAdd is to either have access to the labels of the attributes introducing bias in the data or to be able to train attribute label predictors on another dataset where these labels are available. Through a comprehensive experimental evaluation, we show that the proposed approach surpasses the state-of-the-art in single- as well as multi-attribute bias scenarios.

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

## A APPENDIX

### A.1 MODEL ARCHITECTURE

In experiments involving the MNIST-based datasets, namely Biased-MNIST and FB-Biased-MNIST, we utilize a simple Convolutional Neural Network (CNN) architecture outlined in (Bahng et al., 2020), which comprises four convolutional layers with $7\times7$ kernels and a classification head. For the experiments involving the Biased-UTKFace, Corrupted-CIFAR10, and CelebA datasets, we adopt the ResNet-18 architecture (He et al., 2016). For Waterbirds and UrbanCars datasets, we use ResNet-50 networks.

### A.2 IMPLEMENTATION DETAILS

We employ the Adam optimizer with a 0.001 initial learning rate, which is divided by 10 every 1/3 of the training epochs. Batch size is fixed at 128 and weight decay is set to $10^{-4}$. Following previous works (Hong & Yang, 2021; Sagawa et al., 2019; Li et al., 2023), we train the models on Biased-MNIST and FB-Biased-MNIST datasets for 80 epochs. For Biased-UTKFace and CelebA, training duration is set to 20 and 40 epochs, respectively. As for the Corrupted-CIFAR10 dataset, models are trained for 100 epochs using a cosine annealing scheduler. For the Waterbirds and UrbanCars datasets, we do not use a learning rate scheduler, and the models are trained for 300 and 100 epochs, respectively. Following the initial training phase, the classification head of all models is fine-tuned for an additional 20 epochs. Regarding the bias-capturing models, for Biased-MNIST, FB-Biased-MNIST, Waterbirds, UrbanCars, and CelebA datasets, they are trained on the same dataset as the main model using the attributes introducing bias as target attributes. For Biased-UTKFace we employ the pretrained bias-capturing classifiers provided by Sarridis et al. (2023a). Finally, for Corrupted-Cifar10 we just project the one-hot vectors representing the texture labels (i.e., the attribute introducing the bias) to the feature space of the main model without using a trainable bias-capturing model. All experiments are conducted on a single NVIDIA RTX-3090 Ti GPU and repeated for 5 different random seeds.

### A.3 ABLATION STUDY

In this section, we explore the ways of integrating bias-capturing features into the training process. Table 11 presents a comparison of BAdd's performance when the bias-capturing features are added to the main features versus when they are concatenated with them. As one may observe, the concatenation approach is much less effective than the addition. This is anticipated, as relying on $\mathbf{b}$, with non-zero corresponding weights, would perform poorly on balanced settings (random background color), while not relying on it, with $\sim 0$ corresponding weights, would be equivalent to the sub-optimal vanilla training. Furthermore, Tab. 12 demonstrates how the selection of layer to incorporate the bias-capturing features affects the performance of BAdd. The penultimate layer yields the most favorable performance, as the shallower the selected layer, the fewer layers remain independent of the protected attributes.

Table 11: Addition vs Concatenation: Biased-MNIST performance comparison between different approaches of integrating bias-capturing features.

| Method | $q$ | | | |
|---|---|---|---|---|
| | 0.99 | 0.995 | 0.997 | 0.999 |
| Concatenation | 91.5 | 81.7 | 70.3 | 36.5 |
| Addition | **98.1** | **97.3** | **96.3** | **91.7** |

Table 12: BAdd performance on Biased-MNIST with $q = 0.99$ when considering different layers for incorporating the bias capturing features.

| Method | Layer | | | |
|---|---|---|---|---|
| | 1st | 2nd | 3rd | 4th |
| BAdd | 74.6 | 85.8 | 97.7 | **98.1** |

Also, we explore how BAdd performs when used on datasets with a very limited degree of bias. To assess this, we utilize the Biased-MNIST dataset with low $q$ values - specifically, 0.1, 0.3, 0.5, and 0.7. As shown in Tab. 13, BAdd maintains model performance consistently (i.e., 99.3%) across all the levels of data bias.

Table 13: BAdd accuracy on fair (i.e., $q = 0.1$) or slightly biased data (i.e., $q = \{0.3, 0.5, 0.7\}$).

| Method | $q$ | | | |
|---|---|---|---|---|
| | 0.1 | 0.3 | 0.5 | 0.7 |
| Vanilla | 0.993 | 0.992 | 0.991 | 0.989 |
| BAdd | 0.993 | 0.993 | 0.993 | 0.993 |

Furthermore, in Section 5, we show that BAdd can be combined with either a trained bias capturing model or a projection of one-hot vectors representing the biased attribute labels to the space of $\mathbf{h}$ for deriving $\mathbf{b}$. When employing a bias-capturing classifier, a deep learning model is specifically trained to predict the protected attribute, such as race, gender, hair color, or background. This process encourages the model to learn richer and more diverse latent features associated with these attributes. By focusing on predicting these protected attributes, the model captures subtle, underlying patterns in the data that may be overlooked if solely relying on labeled attributes. Such comprehensive representations can improve the model's understanding of complex visual features, ultimately enhancing the efficacy of bias mitigation. Conversely, the approach of projecting one-hot encoded labels is computationally less intensive and easier to implement as it does not require any additional training steps. However, this method may not effectively capture the intricate visual features that a dedicated bias-capturing classifier can uncover. Table 14 and Tab. 15 provide a comparison of these two BAdd variants on Biased-MNIST and Biased-UTKFace datasets, respectively. The flexibility in choosing between these approaches allows practitioners to balance implementation simplicity with the richness of feature representation. Utilizing a trained bias-capturing model may lead to more effective bias mitigation, especially in datasets where the complexity of visual features plays a critical role.

Table 14: Bias capturing model vs projection: Performance on Biased-MNIST.

| Method | $q$ | | | |
|---|---|---|---|---|
| | 0.99 | 0.995 | 0.997 | 0.999 |
| Vanilla | $90.8_{\pm 0.3}$ | $79.5_{\pm 0.1}$ | $62.5_{\pm 2.9}$ | $11.8_{\pm 0.7}$ |
| BAdd w/ projection | $97.4_{\pm 0.2}$ | $94.8_{\pm 0.6}$ | $90.1_{\pm 1.7}$ | $65.4_{\pm 4.4}$ |
| BAdd w/ bias capturing model | $\mathbf{98.1_{\pm 0.2}}$ | $\mathbf{97.3_{\pm 0.2}}$ | $\mathbf{96.3_{\pm 0.2}}$ | $\mathbf{91.7_{\pm 0.6}}$ |

Table 15: Bias capturing model vs projection: Performance on Biased-UTKFace.

| Method | Bias | | | |
|---|---|---|---|---|
| | Race | | Age | |
| | Unbiased | Bias-conflicting | Unbiased | Bias-conflicting |
| Vanilla | $87.4_{\pm 0.3}$ | $79.1_{\pm 0.3}$ | $72.3_{\pm 0.3}$ | $46.5_{\pm 0.2}$ |
| BAdd w/ projection | $89.7_{\pm 2.6}$ | $88.7_{\pm 4.5}$ | $78.3_{\pm 1.1}$ | $61.8_{\pm 3.1}$ |
| BAdd w/ bias capturing model | $\mathbf{92.2_{\pm 0.2}}$ | $\mathbf{93.3_{\pm 0.2}}$ | $\mathbf{80.3_{\pm 0.8}}$ | $\mathbf{73.6_{\pm 1.0}}$ |

Moreover, Tab. 16 reports the performance of BAdd with several widely adopted backbone architectures, including ResNet-18 (He et al., 2016), EfficientNet-B0 (Tan & Le, 2019), Swin Transformer-Tiny (Liu et al., 2021b), and ViT-Base-Patch16-224 (Dosovitskiy, 2020). The results demonstrate the effectiveness of BAddon both CNN-based and transformer-based architectures. Furthermore, we explore scenarios where bias labels are unreliable. Specifically, Tab. 17 reports the performance of BAdd under different error levels in the protected attribute annotations. Rather than utilizing a bias-capturing model, we adopt label projection, which allows for the controlled injection of label errors. The results demonstrate that even with significant levels of annotation errors (up to 40%, noting that a 50% error rate would correspond to random classification in binary tasks), BAdd consistently outperforms the baseline model, which achieves unbiased and bias-conflicting accuracies of 87.4% and 79.1%, respectively. Finally, Tab. 18 illustrates that BAdd exhibits minimal sensitivity to the choice of batch size.

Table 16: Results on UTKFace for different architectures.

| Method | Bias: Race | |
| --- | --- | --- |
| | Unbiased | Bias-conflicting |
| ResNet-18 | 92.24 | 93.33 |
| EfficientNet-B0 | 91.89 | 90.97 |
| Swin Transformer-Tiny | 92.35 | 92.12 |
| ViT-Base-Patch16-224 | 92.44 | 93.49 |

Table 17: Bias-labels reliability: Performance on UTKFace with varying bias-labels error levels.

| Bias-labels Error | Bias: Race | |
| --- | --- | --- |
| | Unbiased | Bias-conflicting |
| 0% | 89.68 | 88.71 |
| 3% | 89.78 | 87.58 |
| 5% | 89.11 | 85.21 |
| 10% | 89.24 | 84.78 |
| 20% | 88.58 | 81.59 |
| 40% | 88.48 | 80.99 |

Table 18: Impact of batch size on performance: Results on UTKFace for different batch sizes.

| Batch Size | Bias: Race | |
| --- | --- | --- |
| | Unbiased | Bias-conflicting |
| 32 | 91.89 | 93.48 |
| 64 | 92.39 | 94.01 |
| 128 | 92.24 | 93.33 |
| 256 | 91.64 | 93.26 |
| 512 | 90.97 | 93.86 |

## A.4 LEARNING DYNAMICS

As demonstrated in the main manuscript, bias-conflicting samples trigger spikes in the gradients of the bias-aligned loss in subsequent training steps. Figure 5 illustrates that these spikes are mitigated when using BAdd, with the gradients of $\mathcal{L}_{\mathcal{A}}$ staying near zero. This is a direct result of the injection of $\mathbf{b}$ into the learning process.

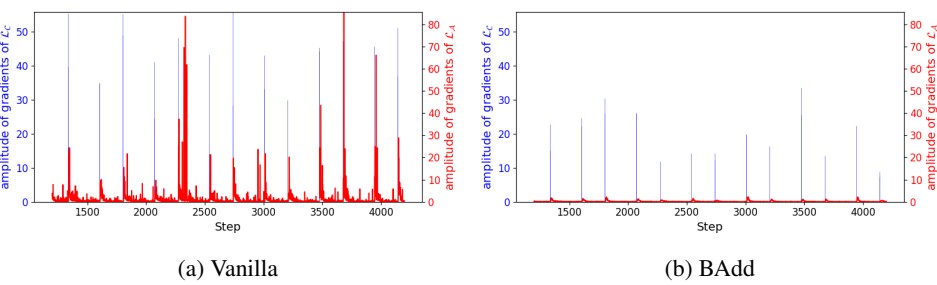

(a) Vanilla

(b) BAdd

Figure 5: BAdd effectively prevents the spikes of the gradients of bias-aligned loss triggered by the bias-conflicting samples of Biased-MNIST. Blue bars indicate the steps where bias-conflicting samples occur, with their height representing the amplitude of gradients.

## A.5    QUALITATIVE RESULTS

Figure 6 visualizes the GradCam activations of a model trained on UrbanCars with BAdd compared to a vanilla model. BAdd effectively swifts the model's focus to the object of interest, with only minor activations in the background that, however, are reflected in the model's performance (i.e., -4.3 BG Gap).

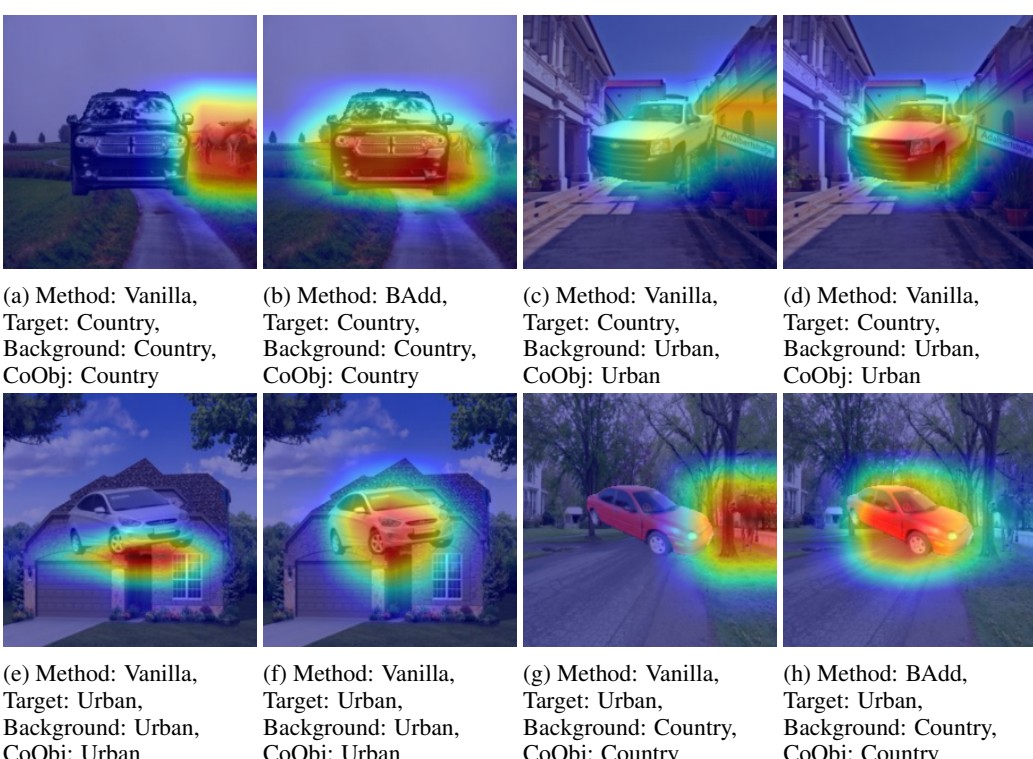

(a) Method: Vanilla, Target: Country, Background: Country, CoObj: Country

(b) Method: BAdd, Target: Country, Background: Country, CoObj: Country

(c) Method: Vanilla, Target: Country, Background: Urban, CoObj: Urban

(d) Method: Vanilla, Target: Country, Background: Urban, CoObj: Urban

(e) Method: Vanilla, Target: Urban, Background: Urban, CoObj: Urban

(f) Method: Vanilla, Target: Urban, Background: Urban, CoObj: Urban

(g) Method: Vanilla, Target: Urban, Background: Country, CoObj: Country

(h) Method: BAdd, Target: Urban, Background: Country, CoObj: Country

Figure 6: Vanilla vs BAdd: GradCam activations on bias-aligned and bias-conflicting samples of UrbanCars dataset.

## A.6    COMPUTATIONAL COMPLEXITY

In this subsection, we discuss the computational complexity of BAdd compared to a baseline (vanilla) model, focusing on both training and inference phases. The conducted analysis assumes a typical setup with a ResNet-18 backbone, input images of size $3 \times 224 \times 224$, and two classes for the biased attribute, similar to datasets such as Biased-CelebA. Figure 7 illustrates the BAdd's training and test phases. It should be stressed that the bias-capturing model is a pretrained model and remains fixed throughout the training of the main model.

The baseline model has a computational cost of 1.818 GFLOPs. When bias-capturing features are added to the penultimate layer of the main model in the BAdd approach, the computational complexity increases slightly by 512 FLOPs, which represents an increase of approximately $2.82 \times 10^{-7}\%$ of the total computational cost. During the fine-tuning phase, where only the final classification layer is updated, the additional computational cost is 1.024 FLOPs, corresponding to an increase of about $5.63 \times 10^{-7}\%$. For the non-trainable components, if label projection is used to extract the bias-capturing features, the additional computational cost is 1024 FLOPs. On the other hand, the computational cost of a bias-capturing model depends on its architecture (which in our case matches the main model). However, it is important to note that this model acts as a feature extractor, so its corresponding features need to be computed only once.

In terms of inference complexity, BAdd introduces no additional computational cost compared to the baseline model, as the bias-capturing component is not utilized during inference. Similarly, the

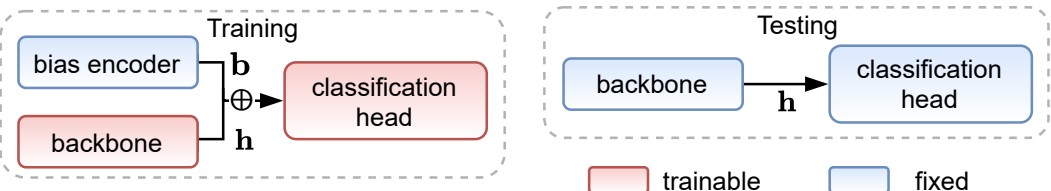

Figure 7: Illustration of BAdd training and test phases.

number of trainable parameters in BAdd is the same as in the baseline model, as the additional components related to the bias features are not trainable. A potential overhead could arise if bias labels or a pretrained bias-capturing model are not available. In such cases, training the bias-capturing model from scratch would add to the overall computational cost. In summary, BAdd introduces minimal overheads during training (i,e., feature addition and fine-tuning the classification layer), while the inference complexity and number of trainable parameters remain equivalent to the baseline.

## A.7 BAdd Requirements

As discussed in Section 6, BAdd requires access to labels (or predicted labels) for the attributes introducing the bias. However, it is important to emphasize that BAdd is more flexible than typical bias-label aware methods. It allows for the use of a bias-capturing model that can be trained on different datasets, making it more adaptable and practical for real-world applications. For example, models processing facial images are often required to avoid biases related to predefined attributes such as race, gender, or age. In these cases, it is straightforward to extract the bias features from existing pretrained models. Additionally, when the specific bias types are unknown, existing bias identification methods can be utilized to infer them (Kim et al., 2024; Zhao et al., 2024). Once the biases are identified, BAdd can be applied to mitigate them. However, exploring bias identification techniques is beyond the scope of this work.

