# OpenReview forum: "BAdd: Bias Mitigation through Bias Addition"
_ICLR.cc/2025/Conference — Submitted to ICLR 2025_

### Official Review · Reviewer_cPAJ · 2024-10-24

**Soundness:** 2
**Presentation:** 2
**Contribution:** 2
**Rating:** 5
**Confidence:** 4

**Summary:**

The paper introduces BAdd (Bias Addition), a method for mitigating bias in deep learning models for computer vision tasks. Bias in datasets often arises from spurious correlations between certain attributes (e.g., gender, race, background) and target variables, leading models to make decisions based on these irrelevant features. Existing bias mitigation methods often struggle in complex, real-world scenarios, especially when multiple biases are present.

BAdd addresses this issue by injecting bias-capturing features into the model's training process. Specifically, it involves adding features that encode the protected (bias-inducing) attributes into the penultimate layer of the model during training. This approach aims to decouple the learning of biased features from the optimization process, allowing the model to learn bias-neutral representations of the data.

The method is evaluated on seven benchmarks, including both artificially biased datasets (e.g., Biased-MNIST, Corrupted-CIFAR10) and more complex, multi-attribute biased datasets (e.g., FB-Biased-MNIST, CelebA, UrbanCars). Notably, it achieves accuracy improvements of +27.5% on FB-Biased-MNIST and +5.5% on CelebA.

**Strengths:**

Versatility: The approach is architecture-agnostic and can be easily integrated into any deep learning model without extensive modifications or pre-processing.
Visualization of Results: Use of activation maps and GradCam visualizations helps to intuitively demonstrate how BAdd shifts the model's focus away from bias-inducing features.

**Weaknesses:**

Assumption of Bias Representability: The method assumes that biases can be captured and represented explicitly. In cases where biases are subtle or unknown, this approach may be less effective.
Potential Increase in Model Complexity: Adding bias-capturing features might increase computational overhead during training and potentially during inference if not properly managed.
Fine-Tuning Requirement: The necessity of a fine-tuning step adds an extra phase to the training process, which might not be ideal in time-constrained or resource-limited applications.
Limited Analysis on Real-World Biases: While the paper includes evaluations on datasets like CelebA, further analysis on real-world datasets with complex, less-defined biases would strengthen the applicability of BAdd.

**Questions:**

Availability of Protected Attributes: How does BAdd perform when protected attribute labels are unavailable or unreliable?

Handling Unknown Biases: In real-world applications where biases may be unknown or multifaceted, how can BAdd be adapted to mitigate biases that are not explicitly identified?

---

> ### Author Response · Authors · 2024-11-16
> **Response to the Reviewer cPAJ**
>
> ## *Responses to the questions*
>
> ### *Response to question 1*
>
> In cases where bias types or labels are unavailable, please refer to our response to reviewer comment iTN8.
> In cases where bias types or labels are unreliable, we present a set of experiments to illustrate the performance of BAdd under varying levels of error in the protected attribute annotations. These experiments are conducted on the UTKFace dataset, where bias is associated with the race attribute. Instead of using a bias-capturing model, we employ label projection (as in Table 15 of the manuscript), enabling the controlled injection of label errors. As shown in the table below, even with significant error levels (up to 40%—noting that 50% error corresponds to a random classifier in binary classification), BAdd achieves higher accuracy than the baseline vanilla model, which shows Unbiased and Bias-conflicting accuracy of 87.4% and 79.1%, respectively.
>
> | Bias-labels Error | Unbiased | Bias-conflicting |
> |-------------------|----------|------------------|
> | 0%                | 89.68    | 88.71            |
> | 3%                | 89.78    | 87.58            |
> | 5%                | 89.11    | 85.21            |
> | 10%               | 89.24    | 84.78            |
> | 20%               | 88.58    | 81.59            |
> | 40%               | 88.48    | 80.99            |
>
> ### *Response to question 2*
>
> In such cases, existing bias identification methods [1, 2] can be employed to infer these biases. Once identified, BAdd can be applied to mitigate them. However, it is worth mentioning that the exploration of bias identification techniques falls outside the scope of this work. For more details on this concern, please refer to our response to reviewer comment iTN8.
>
>
>
> ## *Responses to the weaknesses*
>
> For weaknesses 1 and 4, please refer to our responses to the questions.
>
> ### *Response to  weaknesses 2 and 3*
>
> Some very limited added complexity occurs only in the training phase of the model. In the inference stage, the bias-capturing part of the framework is not involved at all. In addition, the fine-tuning step does not require additional resources (i.e., GPU VRAM), as it is applied after the regular training phase and involves only one trainable layer (i.e., the classification layer). For more details on the complexity of the proposed method please refer to the corresponding answer to the reviewer e7ck.
>
>
> [1] Kim, Y., Mo, S., Kim, M., Lee, K., Lee, J., & Shin, J. (2024). Discovering and Mitigating Visual Biases through Keyword Explanation. In Proceedings of the IEEE/CVF Conference on Computer Vision and Pattern Recognition (pp. 11082-11092).
>
> [2] Zhao, Z., Kumano, S., & Yamasaki, T. (2024). Language-guided Detection and Mitigation of Unknown Dataset Bias. arXiv preprint arXiv:2406.02889.

---

> ### Author Response · Authors · 2024-11-20
>
> Dear reviewer,
>
> We once again thank you for your valuable feedback. Given the short time-frame of the discussion period, we would be grateful if you could let us know if our responses have addressed your concerns as well as if you have any further comments regarding the paper.

---

### Official Review · Reviewer_e7ck · 2024-10-29

**Soundness:** 3
**Presentation:** 3
**Contribution:** 2
**Rating:** 8
**Confidence:** 4

**Summary:**

The paper introduces a BAdd approach to mitigate the effects of biased data during model training. The idea is to incorporate captured bias features into the final layer of the model, which helps the model be invariant for these features and create a bias-neutral feature representation.

**Strengths:**

1-	The problem is described clearly.

2-	The investigated problem is important.

**Weaknesses:**

1-	The BAdd approach seems similar to FLAC [1].

2-	On page 3, The author claims that BAdd is easily applied to any network architecture and to any CV dataset. This claim needs to be proven.

3-	The paper does not discuss the computational complexity or scalability of the proposed approach in detail, which could be a concern for large-scale applications.

4-	Limited Ablation Studies: While the paper includes some ablation studies, more extensive ablations could strengthen the claims about the individual contributions of each component of the proposed method.

[1] Ioannis Sarridis, Christos Koutlis, Symeon Papadopoulos, and Christos Diou. Flac: Fairnessaware representation learning by suppressing attribute-class associations. arXiv preprint arXiv:2304.14252, 2023a.

**Questions:**

1-Describe the key differences between the BAdd and  FLAC [1].

2-Conduct experiments using different architectures, such as ViT with BAdd, to prove that the approach works properly with different architectures. Also, use the ImageNet dataset to show the approach performance with the balanced dataset.

3-Discuss the computational complexity of BAdd approach.

4-We encourage the author to do more ablation studies on the proposed approach to show consistency, such as changing the batch size.

---

> ### Author Response · Authors · 2024-11-16
> **Response to the Reviewer e7ck**
>
> 1. While both BAdd and FLAC aim at learning bias-neutral representations, their approaches to bias mitigation are fundamentally different:
>    * FLAC introduces a sampling mechanism to identify bias-aligned and bias-conflict samples. It then employs a loss function that encourages the main model to exhibit high or low similarities between samples based on the similarity levels observed in the bias-capturing representations.
>    * In contrast, BAdd introduces a mechanism that prevents bias from being introduced to the model by injecting certain features relevant to the bias within the training phase of the model.
>
>    More intuitively, FLAC (as well as most of the relevant methods) tries to penalize the bias that is already introduced in the model, while BAdd is a proactive bias mitigation method that intervenes earlier in the training process, addressing the root cause of bias propagation within the model itself.
> 2. Please find the additional experiments concerning 3 additional networks below.
>
>    *Table: Results for UTKFace (race).*
>    |Model|Unbiased|Bias-conflicting|
>    |-|-|-|
>    |ResNet18|92.24| 93.33|
>    |EfficientNet-B0|91.89| 90.97|
>    |Swin Transformer-Tiny|92.35| 92.12|
>    |ViT-Base-Patch16-224|92.44|93.49|
>    Furthermore, regarding the evaluation of the proposed method on balanced or unbiased data, we have already a relevant set of experiments presented in Table 13, where we evaluate BAdd’s performance for different bias levels, including balanced data (i.e., q=0.1).
> 3. To address the reviewer's comment regarding the computational complexity of the BAdd approach, we provide a detailed analysis comparing BAdd to the baseline (vanilla) model for a typical example involving a ResNet18 network, input images of size 3x224x224, and two classes for the biased attribute (similar to CelebA or UTKFace experiments).
>
>    *FLOPs Analysis for Trainable Components*:
>    - **Baseline Model**: The baseline model's computational cost is 1.818 GFLOPs.
>    - **BAdd (Bias Feature Addition)**: The addition of bias-capturing features into the main model’s penultimate layer increases the computational complexity by 2.82e-7% (i.e., +512 FLOPs).
>    - **BAdd Fine-Tuning Phase**: The fine-tuning process, which involves updating only the final classification layer, incurs an additional computational cost of 5.63e-7% (i.e., +1.024 FLOPs). This phase remains efficient as only a small number of parameters are updated.
>
>    *FLOPs Analysis for Inference Components*:
>    - **Projection**: If projection is employed for extracting bias features, the computational cost is only 1024 FLOPs.
>    - **Bias-Capturing Model**: The computational cost of the bias-capturing model depends on its architecture, which in this case matches the main model. However, it is important to note that this model acts as a feature extractor, so its corresponding features need to be computed only once.
>
>    *Inference Complexity*:
>    The inference time of the BAdd approach is equivalent to the baseline, as the bias-capturing component is not involved during inference.
>
>    *Number of Trainable Parameters*:
>    Similarly, the number of trainable parameters in BAdd is the same as in the baseline model, as the additional components related to the bias features are not trainable.
>
>    *Potential Overheads*:
>    The main potential overhead arises if bias-labels are unavailable or if there is no pretrained bias-capturing model. In such cases, training a bias-capturing model from scratch would add to the overall computational effort.
>
>    In summary, BAdd introduces minimal overheads during training (e.g., feature addition and fine-tuning the classification layer), while the inference complexity and number of trainable parameters remain equivalent to the baseline.
>
> 4. We appreciate the reviewer's suggestion to conduct additional ablation studies. First, we would like to highlight that Tables 11-14 involve ablation studies that sufficiently cover the main components that could potentially affect the performance of the proposed approach. Additionally, we would like to stress that in this domain, ablations involving standard hyperparameters such as batch size or optimizer types are not commonly performed due to their limited contribution to understanding the method's core behavior. Nonetheless, we have conducted a relevant experiment to assess the effect of varying the batch size and found consistent performance across different configurations (see Table below). We are open to performing further experiments if they add meaningful insights into the behavior and performance of our approach.
>
>
>
>    *Table: Results for UTKFace (race).*
>    | Batch Size | Unbiased | Bias-conflicting |
>    |-|-|-|
>    |32| 91.89| 93.48 |
>    |64| 92.39| 94.01|
>    |128| 92.24| 93.33|
>    |256| 91.64| 93.26|
>    |512| 90.97| 93.86|

---

> ### Author Response · Authors · 2024-11-20
>
> Dear reviewer,
>
> We once again thank you for your valuable feedback. Given the short time-frame of the discussion period, we would be grateful if you could let us know if our responses have addressed your concerns as well as if you have any further comments regarding the paper.

---

> > ### Comment · Reviewer_e7ck · 2024-11-20
> >
> > Thank you for the detailed responses, clarifications, and additional experiments. Most of my concerns have been addressed.

---

> > > ### Author Response · Authors · 2024-11-21
> > >
> > > We are glad to have addressed your concerns and would greatly appreciate it if you could revise your rating accordingly.
> > >
> > > We remain open to any open issues you may have.

---

### Official Review · Reviewer_JPua · 2024-11-03

**Soundness:** 3
**Presentation:** 3
**Contribution:** 4
**Rating:** 6
**Confidence:** 4

**Summary:**

The model proposed in this paper learns unbiased features by adding specific elements that capture potential biases during training. This ensures that the model doesn’t rely on biased information when making predictions. BAdd was tested on various datasets, including those with single and multiple biases, and showed strong improvements in reducing bias and overall model performance.
So,  this paper:
1.Introduced BAdd, an easy and effective method for learning unbiased features by incorporating bias-detecting elements into training.
2.Evaluated BAdd on several benchmarks, including different types of biased datasets, showing that it outperforms other state-of-the-art methods.

**Strengths:**

1.BAdd reduces bias effectively by adding bias-related features into the training, helping the model avoid being influenced by biased data.
2.The authors showed that BAdd works well on different datasets, with consistent improvements in various bias situations, proving that the method is scalable and works in different applications.

**Weaknesses:**

1.BAdd requires a classifier or labels that identify the bias, which may not always be available. The authors could consider ways to detect and handle biases automatically without needing predefined labels.
2.The method mainly addresses visual biases, and it’s unclear if it works for other types of biases, like those in text, limiting its use beyond visual data.
3.The paper is clear, but lacks a detailed comparison to standard deep learning training. It’s not explained how the bias-detecting classifier fits into the training process.
4.In line 728, the tense is inconsistent—“were” should be changed to “are.”

**Questions:**

please check the weakness part, and give some explanations.

---

> ### Author Response · Authors · 2024-11-14
> **Response to the Reviewer JPua**
>
> 1. In such cases, existing bias identification methods [1, 2] can be employed to infer these biases. Once identified, BAdd can be applied to mitigate them. However, it is worth mentioning that the exploration of bias identification techniques falls outside the scope of this work. For more details on this concern, please refer to our response to reviewer iTN8.
>
> 2. BAdd is indeed tailored to visual data. However, based on the theoretical analysis provided, BAdd could potentially be applied to any context involving a feature extractor (e.g., backbone) and a classifier. While exploring its applicability to other data modalities is beyond the scope of this work, we are open to conducting further experiments on other types of data in the camera-ready version of the paper, given the limited timeline of the discussion period.
>
> 3. To clarify, the bias-capturing model is not trained concurrently with the main model; rather, it serves as a bias feature extractor. The main model is trained using standard deep learning practices, and the bias-capturing model provides features that are used to ensure that biases do not influence the final learned representations. We plan to include a relevant figure and a more detailed description in the paper to make this clearer to the reader.
>
> 4. Thank you for pointing that out.
>
> [1] Kim, Y., Mo, S., Kim, M., Lee, K., Lee, J., & Shin, J. (2024). Discovering and Mitigating Visual Biases through Keyword Explanation. In Proceedings of the IEEE/CVF Conference on Computer Vision and Pattern Recognition (pp. 11082-11092).
>
> [2] Zhao, Z., Kumano, S., & Yamasaki, T. (2024). Language-guided Detection and Mitigation of Unknown Dataset Bias. arXiv preprint arXiv:2406.02889.

---

> ### Author Response · Authors · 2024-11-20
>
> Dear reviewer,
>
> We once again thank you for your valuable feedback. Given the short time-frame of the discussion period, we would be grateful if you could let us know if our responses have addressed your concerns as well as if you have any further comments regarding the paper.

---

### Official Review · Reviewer_iTN8 · 2024-11-03

**Soundness:** 2
**Presentation:** 3
**Contribution:** 2
**Rating:** 5
**Confidence:** 3

**Summary:**

The document introduces BAdd, a method for mitigating bias in deep learning models by injecting bias-capturing features into the training process. This approach aims to create bias-neutral representations by incorporating features that encode bias-inducing attributes, thus preventing the model from relying on these biases during training.


The core idea is to divide the training loss between bias-aligned and bias-conflicting samples and study their behavior. To mitigate the loss spike due to underfitting on bias-aligned samples, the authors introduce the attributes themselves as features in the final layer, which mitigates the loss spikes and leads to proper training.

**Strengths:**

- The paper is clear and well-written.

- The core idea seems interesting and is supported by various experiments.

**Weaknesses:**

The idea of Bias Injection to Mitigate Bias has been used in the following work [1].

Although the idea is that a bias injection module, can prevent the loss spike. However, when the loss is forced to be zero, it needs to overcorrect the bias injection module, does it lead to correct features?

For the bias-aligned examples, the network can probably take the shortcut. Hence, the learning needs to happen just which Bias Corrected samples. In case B_c >> B_a won’t it affect the learning of diverse features? It would be great if the authors could clarify this aspect more.


The introduction section mentions that the method is more suitable for real-world. However, on closely examining the method, I found that BAdd also requires knowing the bias attributes. Could the authors please clarify on this aspect?

**Questions:**

Please see the questions in the weakness section.

---

> ### Author Response · Authors · 2024-11-14
> **Response to Reviewer iTN8**
>
> 1. It seems the citation [1] was referenced without the actual paper being provided. Could you please share the complete reference for this work? This will help us include it in the related works section and discuss its differences compared to our proposed method.
>
> 2. Loss spikes occur when the main model attempts to learn the correct features, as the loss temporarily increases for bias-aligned samples, which make up the majority of the training data. Injecting bias-capturing features helps prevent these loss spikes by ensuring the model continues to make correct predictions on these samples, leveraging the inherent biases in the data. This means that the bias-capturing features do not contribute to an increase in the loss; rather, they help reduce it. This allows the main model to explore other diverse features (i.e., the correct features) that further minimize the overall loss, without needing to “correct” any errors introduced by the bias-capturing module.
>
>    For bias-conflicting samples, which represent only a small fraction of the training data, the bias-capturing features do contribute to the loss. Although this contribution is minimal due to the small number of such samples, it aligns with our goal of bias mitigation, as it prompts the main model to place greater emphasis on these samples, requiring additional effort to shift predictions in the correct direction despite the noise introduced by the bias features. In addition, it is important to note that “forcing zero loss” is an edge case and does not typically occur during deep learning model training.
>
>    We believe that both theoretical analysis and experimental results presented in the manuscript support the described behavior but we are open to incorporate concrete suggestions if these help make our contribution clearer.
>
>
> 3. In a biased dataset, bias-aligned samples are by definition more (in number) than bias-conflicting samples. If $\mathcal{B}_c≫\mathcal{B}_a$, this implies that $\mathcal{B}_c$ represents the bias-aligned examples, while $\mathcal{B}_a$ represents the bias-conflicting examples. However, the reviewer may be suggesting an investigation into BAdd's performance across varying levels of dataset bias (i.e., the ratio between $\mathcal{B}_c$ and $\mathcal{B}_a$). Such an analysis is provided in the Appendix (see Table 13) and shows that BAdd demonstrates consistent performance across all the levels of data bias.
>
> 4. Indeed, BAdd requires knowledge of bias attributes, which is acknowledged as a limitation in the conclusion section of the paper. However, we would like to stress that the core contribution of the method lies in its ability to address multi-attribute biases, which remains a significant challenge in the field.
>
>    To better contextualize this, we can divide the existing literature into four broad categories. First, there are bias-label aware methods for single-attribute biases, where many effective approaches exist. Second, there are bias-label unaware methods for single-attribute biases, where many approaches are available, but they tend to be less effective than those in the first category. Third, there are only a few bias-label aware methods for multi-attribute biases, which remain an open challenge as the few available bias-label aware methods have not sufficiently solved the challenge. Finally, there are bias-label unaware methods for multi-attribute biases, which currently offer no effective solutions. This categorization highlights the difficulty in handling multi-attribute biases, underscoring the importance of BAdd’s contribution.
>
>    Furthermore, it should be stressed that BAdd is more flexible than typical bias-label aware methods, as it allows for the use of a bias-capturing model that can be trained on different data, thus making it more practical to apply in real-world scenarios. For instance, models processing facial images are often required not to exhibit biases with respect to certain predefined attributes, such as race, gender, or age. In such cases, it is straightforward to extract the bias features from existing pretrained models.
>    Finally, in situations where even the bias types are unknown, existing bias identification methods can be employed to infer them [1,2]. Once the biases are identified, BAdd can be applied to mitigate them. However, the exploration of bias identification techniques is beyond the scope of this work.
>
>    We hope this clarification addresses the reviewer’s concerns, and we believe that BAdd makes a meaningful contribution to the problem of multi-attribute bias mitigation in deep learning models.
>
> [1] Kim, Y., Mo, S., Kim, M., Lee, K., Lee, J., & Shin, J. (2024). Discovering and Mitigating Visual Biases through Keyword Explanation. In Proceedings of the IEEE/CVF CVPR (pp. 11082-11092).
>
> [2] Zhao, Z., Kumano, S., & Yamasaki, T. (2024). Language-guided Detection and Mitigation of Unknown Dataset Bias. arXiv preprint arXiv:2406.02889.

---

> ### Author Response · Authors · 2024-11-20
>
> Dear reviewer,
>
> We once again thank you for your valuable feedback. Given the short time-frame of the discussion period, we would be grateful if you could let us know if our responses have addressed your concerns as well as if you have any further comments regarding the paper.

---

### Author Response · Authors · 2024-11-23

Dear Reviewers,

We sincerely hope our responses have clarified your concerns. We understand that the discussion period may be a busy time for you, but we wanted to kindly follow up to check if there are any further points you would like us to address.

---

### Author Response · Authors · 2024-12-02

Dear Reviewers,

We would like to inform you that we have uploaded a revised version of the manuscript based on your valuable feedback.

We would be grateful if you could let us know if there are any remaining points or additional concerns that we should address during this discussion period.

Thank you once again for your time and effort in reviewing our work.

---

### Author Response · Authors · 2024-12-04
**Summary of the reviewers' feedback and our corresponding responses**

We sincerely thank the reviewers for the time and effort they dedicated to reviewing our work. We are confident that our responses have sufficiently addressed the concerns raised. Below, we provide a concise summary of the reviewers' feedback and our corresponding responses:

**Reviewer iTN8** requested clarification on the concept of bias injection and the role of bias intensity in the data. In our response, we provided a detailed explanation of how our method facilitates learning unbiased features and showed that BAdd achieves consistent performance across varying levels of data bias, as illustrated in Table 13 of the paper's appendix.

**Reviewers iTN8, JPua, and cPAJ** raised concerns about BAdd’s reliance on a classifier or labels for bias-related attributes. To address this, we first presented a taxonomy of relevant literature, categorizing methods based on bias-label awareness and the number of biases addressed. We highlighted that mitigating multiple biases is an open and very challenging problem, even for bias-label aware approaches. Additionally, we emphasized BAdd’s flexibility compared to other methods due to its use of a bias-capturing model. Furthermore, while bias identification is beyond the scope of this work, we discussed how such methods could be integrated with BAdd to infer bias labels when they are unknown. Finally, we conducted new experiments to assess how the reliability of bias labels impacts BAdd’s performance. The results showed that BAdd maintains competitive performance even when the bias labels are of low quality.
Reviewer JPua inquired about the applicability of BAdd to non-visual data. Our response outlined the technical requirements for applying BAdd to other modalities, while noting that this work specifically focuses on visual biases. Nonetheless, we expressed our willingness to include baseline experiments in the camera-ready version to inspire researchers in other domains to employ BAdd for their preferred data modalities.

**Reviewer JPua** asked for clarification on integrating the bias-detecting classifier into the training process. In our response, we explained that the bias-capturing model functions as a feature extractor, independent of the main model’s training. To enhance clarity, we included a new figure (Figure 7) in the revised paper illustrating the training pipeline.

**Reviewer e7ck** requested a detailed comparison between BAdd and FLAC. In our response, we provided a thorough description of both methods, highlighting their fundamental differences.

**Reviewer e7ck** suggested testing BAdd using additional architectures. Our response involved experiments with various architectures, including CNNs and transformers, demonstrating BAdd’s consistent performance.

**Reviewers e7ck and cPAJ** asked for a computational complexity analysis. In our response, we presented a detailed analysis of FLOPs, showing minimal overhead during training and equivalent inference costs compared to baseline (biased) models.

**Reviewer e7ck** requested experiments analyzing the impact of batch size on BAdd’s performance. We conducted these experiments and included results demonstrating that BAdd maintains consistent performance across varying batch sizes.

**Reviewer JPua** pointed out a minor inconsistency in the text (line 728), which we corrected in the revised version.

---

### Meta-Review · Area_Chair_Pces · 2024-12-22

**Metareview:**

This paper received mixed reviews. The reviewers appreciated the interesting and model-agnostic approach to debiased training, its consistent improvement across multiple datasets, and extensive experiments and analyses. However, they at the same time raised concerns with the demand for manually annotated (well-represented) bias attributes for training data (iTN8, JPua, cPAJ), incremental novelty (iTN8, e7ck), a potential over-correction issue caused by the proposed method (iTN8), lack of empirical validation for the versatility of the method (e7ck), additional space-time complexity imposed by the method (e7ck, cPAJ), limited ablation study (e7ck), limited analysis on real-world biased data (cPAJ), and some clarity issues (JPua).

The authors' rebuttal and subsequent responses in the discussion period address some of these concerns but failed to fully assuage all of them. After the discussion period, Reviewer iTN8 still had concerns about the manual annotation of multiple bias attributes for training data and the potential technical issue of the proposed method; the former was pointed out by other two reviewers too, they did not come back for discussion though. The AC agrees with the authors that the manual multi-bias labeling can be execused regarding the fact that the task of mitigating multiple biases is extremely challenging. However, the authors are strongly encouraged to at least report results with existing bias discovery methods (e.g., [M2]) to demonstrate that the proposed method is ready to be applied to real-world complex scenarios as the authors argued in the paper; this is also essential since a few prior methods mitigate multiple biases with no manual bias labels [M4, M5], and because these methods exist, someone might think that the authors' claim in the rebuttal are overstated. Moreover, considering that the paper strongly emphasizes the ability of the proposed method to mitigate multiple biases as its contribution, experiments with multiple biases need to be strengthened. Currently, the datasets used for the purpose, except the CelebA variant, are all artifical (relevant to the concern of Reviewer cPAJ on limited analysis on real-world biased data), and details of the CelebA variant (e.g., the number of bias attributes and their populations) have not been reported in either the main paper or the appendix although such details are crucial to examine if the benchmark is valid. In this context, the AC strongly encourages the authors to adopt existing benchmarks and/or evaluation protocols of previous work on mitigating multiple biases [M1, M2, M3] in addition to the UrbanCar dataset; of course, the novelty and contribution of the proposed method compared to the all aforementioned papers [M1, M2, M3, M4, M5] should be clearly discussed in the paper, while the current version does not.

Putting these together, the AC considers that the remaining concerns outweigh the positive comments and the rebuttal, and thus regrets to recommend rejection. The authors are encouraged to revise the paper with the comments by the reviewers and the AC, and submit to an upcoming conference.


[M1] Men Also Do Laundry: Multi-Attribute Bias Amplification, ICML 2023

[M2] Discover and Mitigate Multiple Biased Subgroups in Image Classifiers, CVPR 2024

[M3] Improving Robustness to Multiple Spurious Correlations by Multi-Objective Optimization, ICML 2024

[M4] ExMap: Leveraging Explainability Heatmaps for Unsupervised Group Robustness to Spurious Correlations, CVPR 2024

[M5] Identifying Spurious Biases Early in Training through the Lens of Simplicity Bias, AISTATS 2024

**Additional Comments On Reviewer Discussion:**

The AC found that all reviewers do not have enough expertise in debiased training / group robustness, and thus that their reviews are not sufficiently useful; they often missed crucial limitations of the paper. For this reason, the final decision was heavily influenced by the AC's subjectivity and prior knowledge.

Here is a summary of the reviewers' major concerns and how they are addressed.

- **Demand for manually annotated (or well representable) bias attributes for training data (iTN8, JPua, cPAJ)**: Reviwer iTN8 was not satisfied by the response and the other two reviewers did not come back for discussion. However, the AC found that, since the paper focuses mainly on the multi-bias setting, this can be execused due to the difficulty of the problem setting. Also, there is no prior work mitigating multiple biases without explicit bias labels. However, the AC believes that the authors could at least report results with existing bias discovery methods although the performance does not look good enough. Furthermore, experiments with multiple biases should be substantially strengthend in multiple aspects--lack of details for the benchmarks, comparisons with outdated methods, etc.
- **Incremental novelty (iTN8, e7ck)**: The two reviewers presented a relevant paper each, "RMLVQA: A Margin Loss Approach For Visual Question Answering with Language Biases, CVPR 2023" (iTN8) and "FLAC: Fairness-Aware Representation Learning by Suppressing Attribute-Class Associations, TPAMI 2024" (e7ck). The rebuttal clearly describes the key differences of the proposed method from these papers.
- **Technical issue on the loss function that potentially happens when the loss is close to zero (iTN8)**: The AC found that this comment is valid. However, the rebuttal failed to assuage this issue since the authors simply argued that such a case (i.e., when loss is close to zero) rarely happens without empirical evidences. This is one of the main objections of the reviewer after the discussion period.
- **Limited analysis on real-world biased data (cPAJ)**: The AC found that this concern has not been addressed successfully. The authors focused only on the issue of manual bias labeling, but this reviewer also pointed out that the datasets used for evaluation in this paper are not of natural images, except CelebA. The AC considers this is a non-trivial issue as the authors claimed in the paper multiple times that the proposed method works well even in real-world complex application settings. The AC sees this argument has not been well supported by experimental results.
- **Limited to visual data (JPua)**: The rebuttal does not appropriately address this concern, but the AC does not weigh this heavily when making the final decision since the authors in the paper explicitly target vision systems. However, it would be nice if the authors present results on text data such as CivilComments-WILDS, as the proposed method is not necessarily limited to visual data.
- **Clarity issues (JPua)**: The rebuttal and revision well addressed these issues.
- **Versatility of the proposed methodhas not been empirically validated (e7ck)**: Well addressed according to the reviewer.
- **Imposing additional space-time complexity (e7ck, cPAJ)**: Well addressed by the rebuttal.
- **Limited ablation study (e7ck)**: Well addressed according to the reviewer.

---

### Decision · Program_Chairs · 2025-01-22

Reject